# Environmental Barriers and Facilitators to Physical Activity among University Students with Physical Disability—A Qualitative Study in Spain

**DOI:** 10.3390/ijerph18020464

**Published:** 2021-01-08

**Authors:** Javier Monforte, Joan Úbeda-Colomer, Miquel Pans, Víctor Pérez-Samaniego, José Devís-Devís

**Affiliations:** 1Department of Sport and Exercise Sciences, Durham University, 42 Old Elvet, Durham DH1 3HN, UK; 2Departament d’Educació Física i Esportiva, Universitat de València, C/Gascó Oliag, 3, 46010 València, Spain; joan.ubeda-colomer@uv.es (J.Ú.-C.); miquel.pans@uv.es (M.P.); victor.m.perez@uv.es (V.P.-S.); jose.devis@uv.es (J.D.-D.)

**Keywords:** sport, exercise, qualitative research, ICF framework, narrative environments

## Abstract

This paper qualitatively examines environmental factors operating as barriers and/or facilitators to participation in physical activity (PA) of people with physical disabilities. Interview data were collected from 27 Spanish university students through the innovative method of two-on-one interviews. Thereafter, data were subject to a flexible thematic analysis. Three themes were constructed: associations; PA practice spaces; and nonhumans. Concepts from different theories were used to generate meaningful interpretations around each theme. Drawing on our results, we offer several reflections and directions. Two key messages can be highlighted. First, environmental barriers and facilitators are context dependent and thus do not precede the circumstances in which they emerge. Second, Spanish universities should work harder to become relevant PA promoting environments for students with physical disability. The knowledge generated in this study is expected to inform the design, implementation and evaluation of programs promoting PA among Spanish university students with physical disability.

## 1. Introduction and Purpose

There are around 22,000 university students with disabilities in Spain. The evidence suggests that disability can negatively impact the physical, psychological and social health of such students. For example, this population is at increased risk of experiencing chronic pain, obesity, diabetes type II, depression and anxiety, isolation, decreased community integration and lower life satisfaction, among other health issues [1]. Many of these issues, however, can be managed through physical activity (PA). It is well documented that physically active people are more likely to cope with stress and pain, have fewer illnesses, better relationships and a higher quality of life than those who are less active. Other benefits of PA for the health and wellbeing of people with disability have been proven [2,3].

Despite the importance of PA for this population, Spanish university students with disabilities are still insufficiently active. In particular, a recent study showed that 72.2% of such students did not meet the World Health Organization recommendation of 75 min/week of vigorous PA, while 80.3% did not meet the recommendation of 150 min/week of moderate PA [4]. Given this level of inactivity, it is crucial to and understand the factors that either prevent or encourage their PA participation.

Barriers are factors preventing or hindering regular and satisfying PA participation, whereas facilitators are factors that enable or make it easier to voluntarily participate in quality-time PA. Barriers and facilitators operate in different but mutually affecting levels, including the personal and the environmental [5]. Given the scope of this journal, the present article focuses on environmental factors. Thus, the overarching purpose of the article is to explore environmental barriers and facilitators experienced by a sample of Spanish university students with disabilities.

According to this purpose, it is essential to clarify what we mean by environmental factors. As different definitions exist through the specialized literature, it is useful to align with a given conceptual framework. Like previous research for example, Reference [6], this study follows the framework of the International Classification of Functioning, Disability and Health (ICF) [7], which indicates that ‘environmental factors make up the physical, social and attitudinal environment in which people live and conduct their lives’ (p. 178). Under this definition, environmental factors broadly include social factors, such as the characteristics of the surrounding human population; specific individuals who provide physical or emotional support; the values, norms and beliefs of others; the built environment; equipment and other materials; and services designed to meet the needs of people with disability, along with the systems and policies that manage such services [6]. Finally, the ICF has suggested that any environmental factor can be a barrier either because of its presence (for example, negative attitudes towards people with disability) or its absence (for example, the lack of PA spaces).

## 2. The Study

Research on barriers and facilitators to PA among people with disabilities has been truly diverse in terms of study outcomes, data reporting (only barriers, only facilitators or both) and perspectives of research (quantitative and qualitative) [8]. From the quantitative perspective, the study of barriers and facilitators has provided important knowledge on the factors related to PA participation. However, this perspective fails to fully recognize the voices of people with disabilities and their ‘multiple truths,’ thus ignoring that factors influencing PA participation in people with disability vary greatly from one individual to another. As Hunt and Papathomas [9] reflected, much quantitative research
isolates factors from the context in which they are experienced. For example, we know pain is a commonly reported barrier to physical activity but we do not know how pain occurs (symptomatic or aggravated from exercise) or what other daily activities it may limit. Besides, deductive questionnaires and surveys limit findings to previously identified barriers. (p. 244)

The gaps in our understanding of particular and contextualized experiences are in part compensated by the knowledge generated via qualitative research. However, qualitative studies on barriers and facilitators have mostly described what people say about their experience of participation in PA. Comparatively, a more in-depth understanding of how people construct their meaning of barriers and facilitators has received less attention. Taking up this challenge, our design approaches barriers and facilitators from the point of view of the individual who experiences them within specific (biographical, psychosocial and material) contexts. Through a qualitative lens, we move from ‘macro’ concerns to a micro-level focus in which attention is paid to rich details and specific instances. However, particularities can lead to general statements that go beyond the focus of the work at hand. Whilst qualitative studies are not generalizable in a statistical sense (whereby statistical calculations are used to justify generalizations of the findings to the wider population), qualitative generalizations may prove impactful for PA policy and practice [10]. As Bekker et al. [11] summarized, ‘qualitative research can explore and incorporate dimensions that are not currently represented in the literature, for better and more influential outcomes’ (p. 898).

The present qualitative study was underpinned by an interpretivist paradigm [12], which assumes a relativist ontology (reality is multiple and mind-dependent) and a subjectivist epistemology (knowledge is socially constructed and subjective). More precisely, we adopted a thick interpretative (as opposed to a descriptive) position [13]. This position allows researchers to give meaning to participants’ experiences beyond that which the participants may be able or willing to attribute to it. As Willig [13] elaborated, emphasizing interpretation means ‘abandoning any truth claims regarding the analytic insights produced, arguing instead that what is being offered is the researcher’s reading of the data’ (p. 9). To cool down any possible reaction against ‘bias,’ see -for example- the work of Ponterotto [12], Sparkes and Smith [14] or Smith and McGannon [15], which explains why interpretivist research cannot and should not be judged through post-positivist logics. 

## 3. Methods

The participants in this study were 27 university students with diverse physical disabilities from different regions of Spain who took part in a wider quantitative-qualitative project on their PA participation and psychosocial and environmental factors influences this behavior [4]. Although our broad research project also includes people with intellectual and sensorial disabilities, different disabilities involve different barriers and experiences, which would be too wide to capture in a single qualitative study. As such, we decided to focus on physical disability only. To cover people with a range of experiences within this group, a stratified strategy of sampling was applied [16]. First, we selected the particular categories or groups of cases that we considered should be purposively included in the final sample, including gender (14 men and 13 women) and age (18 to 69 years; M = 40.29, SD = 12.22). We then divided or stratified the sample according to such categories and allocated a target number of participants to each one. While some participants were not concerned about issues of anonymity, others were adamant about preserving it and asked us to not share information that might lead them to be identified. Against this, we resolved to omit demographics, exclude quotes that might reveal too much from those participants showing reservations and use pseudonyms for each participant. 

Ethical approval for this study was obtained from the Ethics Committee of the University of Valencia [H1436947544660] and the informants gave their written informed consent prior to participation. Qualitative data were collected through in-depth, face-to-face interviews. An interview guide was developed and used flexibly as an element of support, rather than a pre-set structure. Cross-sectional, retrospective and projective questions were asked throughout. With a few exceptions, the participants were not directly asked about barriers and facilitators but rather about their personal experiences, circumstances, desires and expectations. This was done to create a narrative space in which barriers and facilitators would emerge as part of the story of the participant and not as a decontextualized and disembodied response.

Both the first and second authors conducted the interviews together at a place of the participant preference. More detailed information on the use of two-on-one interviews can be found in Monforte & Úbeda-Colomer [17] but let us summarize the process. Before starting, ethical and methodological potentials and the perils of using two interviewers were discussed by the research team. An interview guide was also elaborated to direct the interactions. At the outset, each interviewer took a different role. One interviewer would follow the interview guide, whereas the other would focus on taking notes, identifying ‘unexpected’ topics and to ask curiosity-driven questions. Two pilot interviews were then conducted to experiment with the roles and uncertainties of the interviewers. During the following data collection phase, ongoing reflections pre- and post-interview helped us to make appropriate adjustments. The practical wisdom and judgment acquired during the process enhanced synchronization amongst the interviewers, as well as their capacity to minimize ethical and procedural risks in the next interviews. In the last interviews, the interview guide lost prominence and the role of the two interviewers became blurred. Overall, this methodological innovation proved useful for establishing rapport and increased our capacity to access relevant data, among other benefits [17]. The duration of the interviews was rather flexible (lasting from 60 to 200 min) depending on the actual involvement between the participants and interviewers. All the interviews were audio-taped and transcribed verbatim.

A thematic analysis of the data was carried out, following the guidelines and reflections of Braun and Clarke [18,19] and Braun, Clarke and Weate [20]. These authors describe thematic analysis as an appropriate technique to identify patterns across a qualitative dataset, which is not particularly tied to a theoretical framework and thus gives researchers great flexibility in how they use it. As a starting point, we established environments and environmental factors as a deductive frame to organize coding and theme development. Within this frame, we moved to an inductive orientation, so that the analysis was directed by the content of the data we collected. We generated succinct labels (i.e., codes), identifying relevant features of the data that might be important to answer which environmental barriers and facilitators were and are perceived by the participants. Then, we examined the codes and collated data to identify significant broader patterns of meaning (i.e., themes). Candidate themes were checked against the dataset and several meetings were organized between the authors to conduct a detailed analysis of each theme, working out their scope and focus, determining their story and deciding on an informative and evocative name for them.

To give sense to and interpret rather than just describe the results, we made use of concepts from a variety of traditions, as opposed to a single, delimited theory. This does not mean analytical anarchism but an engagement with the well-known image of methodological bricolage [21]. As ‘bricoleurs,’ we selected and purposely used the conceptual resources available to complete *an* analysis. Since the background knowledge and theoretical perspective of some of us differed, disagreements over the interpretation of data occurred, which forced further reflection. Ruminations went on until we arrived at a practical and scholarly satisfactory endpoint for all of us. Critical friends with expertise in disability, PA and qualitative methods were consulted during the analytical process to support informed and disciplined choices (see acknowledgements). 

## 4. Results

The participants in this study mostly concurred that environments played, keep playing and will play a great influence on their PA (non)participation. They perceived that problems in the environment created obstacles to engagement in PA and affected their motivation to participate in different activities. As Lluís reflected, ‘the environment disables you. Even if you have a small disability it makes it bigger, because it doesn’t go with you.’ Both the most active and the most inactive participants indicated the significance of having environmental facilitators. Representative of this appraisal are the words of Itziar: ‘I have to find facilitators, the contexts, the environmental factors that allow me a better quality of life (…) I try to adapt but I cannot fully adapt. I need external help (…) it is important to have a mutual adaptation with the environment.’

Against this general background, the following results are organized around three relevant themes: Associations; PA practice spaces; and nonhumans. These are interconnected and, altogether, capture key environmental factors influencing the PA participation of our sample.

### 4.1. Associations

Most of the participants were involved in associations of and for persons with disabilities. Typically, these associations are labelled with the name of a particular syndrome, impairment or disability, fostering the encounter of persons with, allegedly, similar needs. During the interviews, the participants often discussed the important role that associations played in their lives.

At the outset, they highlighted the value of associations for obtaining information about technical aids and specific features of their impairment. Some stayed there. For example, Naira said that members of her association were ‘a reference group… but only for technical aspects, nothing else.’ Other participants went further to add that their associations provide information on PA opportunities or even offer PA initiatives themselves, including ‘aquatic therapy’ or ‘sporting activities.’ This provision of activities facilitated members to jump into PA. As the quote of Noa illustrates:
In the spina bifida association they set up a wheelchair basketball school and one of the physiotherapists asked me why don’t you try? Back then the chair was… I heard talking about the chair and I started crying. But as I had always liked basketball I went [to the basketball school], it hooked me and I stopped crying about the chair and so.

Exceptionally, some associations were specifically focused on PA. This is the case of Active Disability (a pseudonym), an association that became an adapted gym designed by and for people with disabilities. Helga was one of its members. She reflected that
creating an environment as that of [Active Disability] is critical, because this generates group union, meeting other people… and know other people makes you want to come back to this place to see that people again, at the same time you are doing exercise. So, I reckon it’s very important to create a good environment and that people are comfortable.

Previous research indicated that having someone to exercise with improves peoples’ readiness or willingness to engage in PA [22], as well as their adherence to a PA setting. Additionally, the connections established in the gym might extend to other settings. When Active Disability had to close, Helga’s peers and monitors encouraged her to keep doing PA. That lead her to join an adapted swimming club and get into popular races (when using a handbike was allowed). As such, Active Disability brought Helga from not thinking about PA to routinely engaging in regular PA.

Moving forward, our results showed that the connections between people within associations were built through the stories they told each other. As Perrier, Smith and Latimer-Cheung [23] and others suggested, storytelling is essential for making meaning of disability and PA, as well as for establishing social relationships between peers, monitors and carers. This is something that, as Caddick, Phoenix and Smith [24] argued, constitutes a key element of psychological wellbeing. Isabel highlighted the flow of stories in her association and compared it to the gym.
For me, the Multiple Sclerosis Association is like my second home. Why? Because I go there and while I wait for someone to finish, I am talking with someone else behind... it is like in the gym, you know? You chat with everybody and you tell them your stories and listen to theirs… and that’s cool.

Following the thoughts of Perrier, Smith and Latimer-Chung [25], associations worked as narrative environments that motivated the participants to engage in PA. Within associations, people might feel they belong to a community that enhances perceptions of social acceptance, self-worth and camaraderie. As the following quote from Mariano illustrates:
For me, joining an association is very healthy. An association is, you know… you meet people who share the same issue, that is, we are all disabled. That leads us to make ties. The association helps you share, you share feelings, you share ... grievances. Either you tell some other how you are. ‘Cause you say “good morning” to everybody when you arrive. “Are you OK?”. “How does it go?”. I find it interesting. It seems important to me. It cheers people up. ‘Cause associations are mainly to help people who do not feel good. Mainly for them…

In this example, stories perform social comparisons, enabling individuals to evaluate themselves in comparison to others. According to Sparkes, Pérez-Samaniego and Smith [26], social comparison processes are performed via storytelling and, therefore, are narratively constructed in essence. If, as stated in the former quote, stories shared in associations are to help those “who do not feel good,” the upward social comparison would predominate. Upward social comparison is an individuals’ better than them to compare themselves with others who are in a more favorable situation and appear to be doing better than them in the particular dimension being compared. In this vein, social comparison can fulfil functions such as ‘providing useful information about where one stands in one’s social world, feeling better about oneself and learning how to adapt to challenging situations’ [27] (p. 16), all of them vital to individuals with disability.

Individuals, however, may also compare themselves with others who are in a less favorable position and appear to be worse off. This is referred to as downward social comparison. For example, Itziar reflected that ‘sport is promoted without conviction. It is all for people that are really bad. The association has not helped me [because] I am not in the realm of those that are hopeless.’ Similarly, Aina said:
Because going to an association like that… you just stay there and listen to everyone’s complaints. It is like: What hurts you? And what about you? Could you move your toe? I can’t, I can’t go and listen to all this crap and how he deals with it.... I have enough with my own problems to pay attention to someone else’s life.

Aina refuses to compare her stories with those she hears in the associations, because they appear to be worse off in relation to her own. In this case, the association provides the environment for stories and narratives she does not want to compare herself to. In most cases, being detached from associations was not an initial position or disposition. Rather, it happened after some time listening to downward stories. In other words, staying away from associations was a product of progressive realizations or what Denzin [28] called cumulative epiphany:
Ok, at first, especially the first years, I really need to go there [to the association], because I wanted to feel identified with a group but then I realized that I don’t have to relate only to people with disabilities, who are also a double-edged sword. Because if you feel bad, if you are always in pain or the prostheses go wrong and so on, it [social contact in associations] would harm me.

Depending on this key factor (i.e., social comparisons), associations can become narrative environments in which the disability identity is reinforced or devalued. When it is reinforced, people stay there and, if the association promotes PA, they are likely to engage in. In contrast, downward social comparisons might limit the opportunities to engage in the PA opportunities that some associations create.

### 4.2. PA Practice Spaces

The participants do or did PA in different settings. The conventional gym was often mentioned, for better and for worse. For some, it was the most appropriate space. Naira, for example, commented that
I spent some time doing it (PA) at home but impossible… The gym is for me the right space. Not the street, because there are barriers… perhaps it’s raining, the road has an irregular route, I can fall… I prefer the gym.

On the other hand, the view of the gym as a trouble-free environment was rare. Some participants noted several problems. Consistent with the specialized literature, participants noted that lack of knowledge from professionals and lack of information about the PA frequency, intensity type and time (FITT) can turn people away from the gym. In this respect, Juanan said that in ‘the gym, the instructors, when you tell them about your impairment, they go blank. The equipment… you don’t know how to use it neither, nor for what.’ Besides, participants expressed that another key barrier to gym use was through experiencing oppressive practices from both the physical structure of the gym and interactions with others in the gym.
There is a group of people [in the gym] that are so aware of the physical aspect, so I try to go unnoticed. Not because I don’t have the tools to go with the prosthesis but I don’t feel like being stared at either, because I want to stay focused [on the exercise routine]. (Juan)

The experience of being stared at by others can have negative effects on people with disabilities [29]. For example, feeling observed might invalidate people with visible physical disabilities based on public perceptions of normality, beauty and perfection. Isabel’s experience shows these consequences in action. She told us that, after starting the university,
It was difficult for me to make a comeback to sport and I started in a gym that I think it was horrible, to be honest, I didn’t like it. I don’t know, maybe because I saw an aspect of sport that I didn’t contemplate in my life, in my story. Which was seeing sport simply for beauty purposes, you know what I mean. For the physical appearance, more superficial. So, I have always been very sensitive towards those things, you know what I mean.

It is worth noting that, in our sample, the perception of being stigmatized and judged was found both in men and women. To put an example, a man called Lluís reflected that
Being stared at the machines bothers me. They shouldn’t disturb, would you like being stared at all the time? It’s not that they look at you, they examine you all the time. I lack one leg. They are looking at me everywhere. You know the feeling of: it bothers me (…) And they ask you, what happened to you, oh sorry, sorry, so unlucky… Ah and you have good luck, don’t you, asshole?

Connections can be drawn here with the analysis of Richardson et al. [30]. Drawing on the results of their qualitative analysis, these authors reflected that disabled bodies in the gym go completely against the aesthetic values the conventional gym aligns to. As their and our results show, failing to match the culturally “normal” body can lead people with disabilities to perceive the gym as unsuitable for them to exercise, regardless of specialized equipment and knowledgeable instructors.

Importantly, most participants that gave up working out in a gym stressed that they would participate if this and other things would change. Aina said that he had the intention to go to the gym but there were ‘7 or 8 steps’ which were ‘a terrible barrier, because what you do is to crawl and try to climb and you destroy yourself… or you don’t go’ (Aina). Similarly, Lluís told that, for him, ‘the problem is to arrive. I always have to go by car. I can park (…) but then I have to walk 300 m and that’s all. It doesn’t work for me.’

Given such accessibility issues, the adapted gym was often the most valued setting. This kind of space has multiple benefits that make it suitable and attractive to people with disabilities, including university students [31]. Speaking about Active disability, Helga said that
Everything was set up with the proper space so that the chairs could move, you know. And this is very important as well, because you see that is a place where they have really thought about people with disability and that they have taken it into account. They haven’t done it because they had to, like the ramps with a sharp end, that are really made, well, because by law you have to put a ramp and you put it that bad but is not really done thinking… for people with disability.

Despite the perceived benefits of adapted gyms, there is few adapted gyms available across the country. This problem is especially acute in small villages, where scarce, if some accessible space is available to do PA. Juanan commented: ‘I lived in a village. The PA opportunities… I had to go to [two Spanish cities], nothing reached me, let’s say that my village was isolated in this respect.’ In the face of this deficiency, some participants opted to ‘assemble a gym at my house,’ which often tried to imitate the functions of the gym: ‘I have one of these big balls, I have the resistance bands, which help me a lot too, they facilitate a lot because you replace the machines of any gym’ (Alba). For Alba and other participants, such a home-made gym is the most suitable option but again, there are a variety of experiences. For example, Laura’s parents transformed their basement into a gym and forced her to do PA to ‘rehabilitate’ all through her girlhood. Now, she considers home-based PA oppressive.

Alongside the gym (be that conventional, adapted or home-based), the swimming pool was identified as a key space for PA participation. Free access to swimming pools provided by regional or local institutions was deemed a key facilitator. Some barriers that hindered or worsened the PA experience were mainly architectural. For example, locker rooms were ‘too far from the pool,’ the toilet was ‘too narrow’ and ‘the chair doesn’t fit.’ Some participants were adamant in that, even when ‘they’ try to build pools accessible, ‘they did that wrong, given their lack of knowledge’ on the needs of people with disability.

Further spaces potentially suitable for PA participation were identified by the participants. For example, the park was important to Núria because ‘[given my condition], I just absorb vitamin D through the sun (…) cycling gave me the vitamin D I need.’ Consistent with the literature [3], ramps too steep and adverse weather conditions were identified as environmental barriers in the outdoors. Akin to a quote from Núria reproduced above, Naira said that ‘The barriers, in my case, are the ramps, the stairs and certain spots where the material is slippery, especially when it rains.’ Most participants perceived that doing PA outdoors ‘would be nice’ but they find hard to imagine trying out without worrying too much about the barriers that they would likely face and would put them in awkward situations.

Remarkably, universities were not deemed relevant environments to do PA. According to the participants, there is a lack of availability of PA resources in their respective universities. With very few exceptions, university services were not deemed as effective platforms to promote PA within or outside the university, in contrast with the potential of associations. Besides, university staff was not believed as key messengers of PA. While not surprising (given the low levels of PA in this population), this general response is a relevant result that will be discussed later.

### 4.3. Nonhumans

The settings considered above were composed of diverse nonhumans (i.e., material things). Our analytical engagement with nonhumans is inspired by sociomateriality, an approach that examines how material elements constitute, mediate and enable social practices by looking at the relational encounters between human and nonhuman actors [32,33].

Our results offer several instances showing that material objects were not passive factors or mere backdrops to participants’ engagement with PA but rather causal actors or mediators. Drawing on a sociomaterial approach called actor-network theory (ANT), Sayes [34] explained that a mediator does more than merely transport action from elsewhere. Namely, it makes a difference; it produces effects and alters situations. It does not just do what anything else would do in its place, but it adds something and it modifies the ‘modes of action’ of other actors -including other objects but also human actors. This idea will be better appreciated in application than in the simple description. Given the purposes of our analysis, we will focus on how objects mediated to facilitate or hinder PA participation. In the case of Núria, for example, diverse objects acted as facilitating mediators, including
shoes, knee pads, anti-inflammatory cream, the rubber ring to sit and the supporting feature for the bicycle seat, because I cannot do it [PA] without it’ (…) [These objects] allowed me to do PA. I wore normal ankle supports but this wasn’t useful and I told him [the orthopedic surgeon]: I need it [PA], because I need to get some sun.

Fluids can also act as mediators. The water of the swimming pool acted as a significant mediator in the case of Itziar, who said: ‘It’s about finding your place (…) where I am functional is in the water.’ Outside the water, ‘I am disabled.’ Itziar felt that water allowed to be herself, not least because PA was ‘part of my life.’ The water, however, needs to have an adequate temperature. Unfortunately, she reflected, there was just one suitable swimming pool with in the area where she lives. And a demand to raise the swimming pool temperature meant a ‘bureaucratic labyrinth’ that hinders the solution of the problem, because ‘nobody seems to listen, nobody tells you anything.’ This situation of administrative silence is shared by other participants. Moreover, when solutions are offered, these are hurried and inadequate. For example, Juanan said that:
There are many sport centers where there is not a specific shower sit, so you have to go and say: hey, I need a chair, I can’t shower. And then they bring you the plastic chair from the cafeteria and then you get by on your own.

In some cases, mediators have an ambivalent role. To illustrate, Naira reflected that her leg prosthesis was part of her and that a reason for doing PA was shaping her muscle so that the prosthesis fits perfectly with her body. At the same time, the fear of breaking a leg due to her prosthesis material restrained her engagement in PA.
You feel fear when doing these exercises lest the material breaks, because I am left exposed. And I am quite relaxed because, all things considered, I have several prostheses. But, what if the person that perhaps is starting has just one? And I make an effort (…) in doing exercise to tone the muscles, especially the quadriceps that is so important to us.

It is also worth mentioning that, with frequency, material objects reflected the desires and expectations of participants in terms of their functional recovery. Some participants adopted restitution to deal with their disability and rehabilitation. Drawing on Frank [35], Monforte, Pérez-Samaniego and Smith [36] explained that restitution is a storyline (i.e., yesterday I was abled, today I am disabled but tomorrow I will be able-bodied again) but also the material elements that perform and make using this storyline possible. In the case of Alba, for example, the stairs constituted restitution and mediated her experience of rehabilitation:
I want to use machines and be able to go up these three stairs. It is like one of these little bridges made of wood for rehabilitation, that you go up one side and down the other. Look, this is a little goal or achievement for anyone. It’s not even an achievement but for me, now, it is, it is a goal: to recover the mobility of the leg.

As a material dimension of restitution, the stairs interact with the hopes and beliefs of Alba, his narrative affinities, his impaired body and other restitution elements that facilitated recovery-oriented PA. Without the stairs, we might speculate, restitution might have not been possible and therefore Alba’s engagement with PA would have been different, if not absent. And, similarly, without restitution, those particular ‘three stairs’ would have not meant an achievement. Following their analysis of restitution in different environments, Monforte, Pérez-Samaniego and Smith [36] argued that PA environments are constituted by material and symbolic elements whose relationship is inseparable.

Sociomateriality allows for exploring how objects and meanings are entangled. The words of Isabel serve as an empirical example of that. Earlier, we showed how she felt uncomfortable in the gym and we related that with normative discourses about the body. Yet, these discourses are materially embedded practices. Isabel said that she ‘felt like there were women all-time looking in the mirror and like (…) I was out of place.’ From this instance, it can be interpreted (from a sociomaterial perspective) that the mirror itself enacts the mirroring body of the gym users -described by Frank [37] as an ideal type of body which defines itself in the image of other bodies around it. Isabel does not talk with those women and she does not know about their body conceptions but despite that she is affected by the idealized materiality of their bodies and their practices towards it, which are directed to mirror a desired, normative body. A body that she, as a disabled person, cannot and do not want to aspire to. She perceives the ableist gaze when the women look at themselves in the mirror and this pushes her away from the environment, in which she feels ‘out of place.’

To deal with the ableist gaze of non-disabled people that surround them while they do PA, students with disability might use objects to hide their disability or pass as able-bodied. Disability passing refers to the way people conceal social markers of impairment to pass as “normal” [38]. For Josep, passing was a necessary condition to do PA in public. He is a colostomized person and uses a fanny pack to cover his colostomy bag for walking and exercising on the beach. He said:
To go to the beach here near Alicante, to any cove. ... then you feel free, you take it easy. You wear a swimsuit or not ... of course, explaining to someone what you’re wearing there [colostomy bag] … Out of question! So I take a fanny pack [to hide his colostomy bag ...] and I spent the whole morning, swimming and so. I need little tricks to keep on doing it [exercise].

The fanny pack plays a mediator role between Josep and his leisure-time PA on the beach. This is because the fanny pack neutralizes his fear of being unmasked as a colostomized individual. From the point of view of critical disability studies, this self-shame can be seen as problematic, as it reveals internalized ableism [39]. Simultaneously, though, passing can be seen productive, at least in this case, because it allows Josep avoiding the glances that constitute a barrier to doing PA in public spaces.

## 5. Some Reflections and Potential Directions

The results of this study support the argument that barriers and facilitators are contingent and dynamic, as opposed to stable and essential. This means that the capacity of a factor to become a barrier or a facilitator is not inherent to this factor but context-dependent. Factors *emerge* as barriers or facilitators (or remain ambivalent) based upon the connections between subjects and the environments in which they become subjects.

In our analysis, we have spotted, selected and explored key factors of a diverse nature (e.g., storytelling, social comparison, object mediation, disability passing). In looking at their influence, we have found diverse and even opposite experiences. Some important factors fostering or hindering participation can be complex, multifaceted and, moreover, what is a facilitator for a person might be a barrier to another. Given such a point, it is fair to suggest that factors need to be contextualized and individualized, rather than fixed in (otherwise useful) lists of barriers and facilitators that overlook much of the complexity of the personal at the expense of the unambiguousness of the statistical.

Certainly, the (hyper)contextual nature of factors complicates the task of PA promotion. If we cannot fix factors, then we cannot provide stable or standardized directions to address them. This does not mean that recommendations are not possible; it means that recommendations ought to consider the relative, fluid and multifaceted nature of barriers and facilitators, as well as problematize universal solutions. A case in point: making PA settings (e.g., gyms) more inclusive is a desirable direction [30] but ‘inclusion through universalization risks a kind of erasure of multiplicity and difference by assuming and (re)producing a predetermined range of bodies and ways of doing-in-the-world’ [40] (p. 1329). Resonating with the work of Shields and Synnot [41] on children with disabilities, the results of our study show that one size does not fit all. Recommendations may be necessary but not enough to understand and guide promotion of PA among university students with disabilities.

Arguably, then, the best way for providers of PA to find out how to include university students with disabilities is to ask them and to observe what they do and need in their everyday life. One scholarly way of doing the above is by conducting ethnographies, that is, by examining how people live, interact and view PA in their daily life. Carrying out an ethnography of a gym, an association or a sports center, would provide the needed contextualized knowledge. To conduct a proper ethnography, you need an ethnographer, that is, a person with ‘trained abilities to look at people, listen to them, think and feel with them, talk with them rather than at them’ [42] (p. 119). This person can immerse in a given setting and provide information about what barriers and facilitators are operating and how the user experience can be improved. Although the figure of the PA ethnographer is virtually non-existent in Spain, we would argue that it will be very beneficial to generate and translate highly contextualized knowledge to managers and practitioners on how to confront barriers, provide facilitators and improve the overall experience of users.

In parallel or as an alternative to employing an ethnographer, education and training should be offered to managers and practitioners on how to effectively spot and tackle barriers and facilitators. One content that could be developed in this training is the use of sensitizing concepts that orient and reorient practice. Concepts can be used as heuristic tools to help thinking through issues that are contingent, fluid and context-sensitive, including barriers and facilitators. To illustrate, the concept of social comparisons used in this paper can draw the attention of health workers to how students with disabilities compared to other people (with or without disability) and which direct and indirect effects have such comparisons in their wellbeing and their PA behavior. Likewise, understanding the concept of ableism and what ableism can cause to people with disability can instigate PA providers to minimize normativity in their practices, adapt spaces such as the toilet, cease to tolerate ableist behavior and expressions from non-disabled users or even work with people with disability towards their empowerment in and through PA, as opposed to (only) functional restoration.

As Goodley [43] suggested, there is often nothing more insightful, useful and powerful than a good concept. Surely, engaging with concepts might be tough but it might equally be extremely rewarding. If professionals come to know a repertoire of useful concepts, they will be in a better position to understand which barriers are operating, for whom, how they are operating, why, with which effects and how could these be removed or transformed into facilitators. As some participants of this study stressed, tackling some barriers requires a considerable economic investment and structural change but with others, it is a matter of being sensitive and see how a particular reality is and could be made otherwise.

Given the highlighted importance of sensitizing concepts for managers and professionals, an interesting line of research is concerned with making theory and concepts judiciously accessible and attractive to more-than-academic audiences. One example of this knowledge translation activity is the work of Smith [44]. Using literary techniques, Smith produced an evocative story that contains evidence and theory on how barriers affect people within a rehabilitation environment. Here, concepts are narrativized in order to make them less abstract and closer to experience. More work in this line should be produced to reach a wider audience of professionals and non-disabled people sharing PA spaces with people with disabilities. A Spanish example in the realm of disability and physical education can be consulted in Martos-García and Monforte [45].

## 6. The Role of Spanish Universities

The participants in this study perceived that, when it comes to promoting PA, universities are highly irrelevant. In the context of this paper, such a result is important enough to give it its own space for discussion.

Our results make apparent the need of reconsidering the role of universities in the promotion of PA amongst this population. In fact, it is argued that they have a great potential to contribute. As a setting in which a large number of students, teaching staff and administration staff participate daily, the university environment constitutes an ideal context for understanding and addressing health challenges. Indeed, this is shown by the several initiatives that have been launched around the world in the last years to enhance the potential of universities in health promotion. Such initiatives are known as ‘healthy universities,’ ‘healthy campuses,’ ‘health-promoting universities’ or similar, depending on the context and they include PA as a fundamental part.

And yet, the potential of healthy universities seems not to be completely unleashed. This is not to say that none of all the students with disability in Spain has benefitted from the university or that each and every Spanish university have done nothing to promote PA amongst the students with disability. We are aware that services, units and individuals exist within some universities and they have developed different initiatives to promote PA. Precisely because of such interest, we argue that universities have the opportunity to take a step forward. In this regard and cautious about the limited scope of our results, we feel responsible for drawing a provocative generalization (Smith, 2018) by making a call for adding universities to the menu of environments for people to do PA and/or get relevant information about how, when and where to do it. Moving forward, universities should facilitate the behavior change journey for a student to go from participating in little or no physical activity to living and enjoying an active lifestyle. Students will not make this behavior change in isolation and, while it is important to give them choice and control in the selection of activities, the supporting role of universities could make a difference.

In this regard, the implementation of PA promotion programs at the university level would be of great relevance. For these programs to be effective, they should be based on the best evidence-based knowledge available. Recent quantitative studies have provided useful knowledge on the barriers to PA experienced by university students with disabilities [46,47]. These studies help to spot what are the sensitive topics that largely affect this population. However, as shown in this paper, we also need to gain awareness of how the meanings of barriers and facilitators are (re)constructed by a highly heterogeneous population. Multi-sector and coordinated interventions need to harmonize these approaches to address the complex, interrelated and contextual effects of facilitators and barriers to PA, while paying due attention to individual needs. To ensure success, closer collaboration between the disability care services of the universities and the sports services of these institutions is also crucial. Finally, this agenda demands effective messaging. In this regard, PA might be promoted within Spanish universities from now on by disseminating the co-produced messages included in the infographic on physical activity for disabled people [48]. This format has been chosen as the preferred one by people with disability and can be easily displayed on a poster, brochure, on digital screens and on social media. Alongside the format of messages, key messengers to promote PA need to be considered. For example, social workers have been identified as key messengers to promote physical activity to and for people with disability [49]. Work is being done to educate and train this group on how to do it safely and effectively (please contact the corresponding author for more information).

## 7. Conclusions

The present article has been developed in accordance with three premises. The first one is that promoting PA is vital to enhance the health of people with disability [1,2,3,49]. The second one is that the removal of environmental barriers coupled with the promotion of facilitating factors is essential for enhancing opportunities for PA and reducing the risk of secondary health conditions in people with disability [2,50]. The third one is that, before any improvements to PA promotion can be instigated, it is necessary to identify such factors and understand how they work and how they can be tackled [51]. With such premises in mind, in this article we have identified and examined barriers and facilitators to PA among Spanish university students with physical disability -a group that we previously identified as inactive [4] in part as a result of environmental barriers [46,47]. To do so, we adopted an interpretive stance [13]. This philosophical stance allows addressing the multiple meanings that people connect to their subjective experiences and interpret the social structures, settings and processes that shape these meanings. Methodologically, we used a qualitative technique called two-to-one interviews in order to generate rich data [17]. Through a thematic analysis of that data [18], we crafted three themes that capture how participants experimented and perceived diverse barriers and facilitators.

The first theme, Associations, illuminate the relative value of associations to promote PA among university students with physical disability. Associations are described as narrative environments [25] that create the conditions to perform social comparisons [26] between members. Upward comparisons facilitate staying in associations, which means more opportunities of engaging in PA. In contrast, downward comparisons lead to disengagement and, in turn, to miss opportunities to get active. In the second theme, PA practice spaces, the spaces in which participants did or do exercise were identified and discussed. Conventional gyms are spaces of interest for people with physical disability but different barriers—that resonate with previous research [e.g., 30]—limit their potential. Adapted and homemade gyms are powerful alternatives. Accessible and acclimatized swimming pools are also enabling spaces. The promotion of PA in open spaces is challenging, as it entails adapting the urban furniture and the construction of inclusive cities. Finally, the theme we called Nonhumans illustrated how material things—be those PA equipment, fluids, disability artifacts or other mundane objects—actively mediate in the PA participation of university students with physical disability. In this regard, we highlight the potential of sociomaterial perspectives [33] for future research on environmental factors.

Taken together, the three themes show that factors are ambivalent and context dependent. This meta-result opens an avenue for a change in the study of factors: if factors are not isolated from the context in which they are experienced, we should study them through context-sensitive approaches and methods, such as ethnographies. This does not mean that conducting ethnographies is the correct way of studying barriers and facilitators or that positivist approaches are not important. It means that generating contextualized knowledge is necessary to advance in our understanding of environmental factors and to improve professional practice.

Finally, the results of this study also drew the attention to the minor role that Spanish universities play in the promotion of health-enhancing PA. On this, the general take-home message is that there is still much work to do to promote PA among people with disability with due confidence. The participants’ perceptions and lifestyles illustrate this. At the same time, their collective voice makes a plea for tackling the environmental barriers that prevent them to participate in PA both within and outside the campus. Once again, they remind us that health is their need and their right. As Juan remarked, ‘the only thing we ask for is a right that belongs to us.’ Universities should listen to and respond to this plea. We hope this paper provides a useful resource to articulate such response.

## Data Availability

The data presented in this study are available on request from the corresponding author. The data are not publicly available due to ethical reasons.

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
