# Peer review of "Environmental Barriers and Facilitators to Physical Activity among University Students with Physical Disability—A Qualitative Study in Spain"

_ijerph, 2021, doi:10.3390/ijerph18020464_

Round 1

Reviewer 1 Report

Very well presented project. I would like a bit more thorough thinking in "conclusions" chapter, but other than that this is very well done.

Author Response

Very well presented project. I would like a bit more thorough thinking in "conclusions" chapter, but other than that this is very well done.

Thank you very much for your comment. As points 5 and 6 discuss the results and present several reflections and recommendations, we decided to produce a condensed conclusion. However, we agree with the reviewer. As such, we have re-written the Conclusions chapter.

Reviewer 2 Report

Peer Review of Manuscript

Environmental barriers and facilitators to physical 3 activity among university students with physical 4 disability: A qualitative study in Spain

Comment

This is a well written article.  Its strength is in the attention to detail particularly around the thoroughness of its methodology.  Of note is the thoroughness of the justification for a qualitative approach.

There is a clear purpose statement although the authors use both goal of this article and purpose of this article.  It is suggested that only one term be used. In my analysis the term should be Purpose.  Furthermore, the heading Introduction, should be renamed as Introduction and Purpose.  

The authors are to be congratulated on the currency of the academic literature. Over 60% of the literature used is post 2012. However, there is a need to thoroughly check the accuracy of the bibliography to ensure all conventions including appropriate punctuation are correct.

The Conclusion is the weakest aspect of the manuscript.  Given the quality of the rest of the manuscript, the conclusion is disappointingly brief and fails to bring the key findings back into focus.  It is suggested that this be re-written to bring the Purpose of the manuscript and its subsequent key findings into focus.  This is an important requirement for acceptance.

While the Conclusion does hint at what this research progresses there is a need to be more focussed on what new knowledge or understanding is progressed. This should be made clearer in the re-write of the Conclusion.

There are 10 keywords listed.  This far too many and the authors need to restrict these to 5.

The Manuscript need to be thoroughly checked for minor grammatical errors and to ensure that sentences are not too long.  As a general rule word count in any sentence should not exceed 40 words.  For instance, the sentence in lines 56-59 is far to long.  There are other instances of this that need to be addressed.  In addition, Line 78 the word relayed should be ‘relied’.

In line lines36 and 135 mention is made of “other benefits” – what are these other benefits.  These need to be specified.

Overall, this is a well-constructed the points raised above. piece of research and is well worth publishing.  However, acceptance is contingent on attending to  

Author Response

There is a clear purpose statement although the authors use both goal of this article and purpose of this article.  It is suggested that only one term be used. In my analysis the term should be Purpose.  Furthermore, the heading Introduction, should be renamed as Introduction and Purpose.  

Thank you very much for your comment. We agree with your observations. We have included the suggested changes in the manuscript.

The authors are to be congratulated on the currency of the academic literature. Over 60% of the literature used is post 2012. However, there is a need to thoroughly check the accuracy of the bibliography to ensure all conventions including appropriate punctuation are correct.

Thank you for your comment. We have checked the accuracy of the bibliography and amended the spotted mistakes.

The Conclusion is the weakest aspect of the manuscript.  Given the quality of the rest of the manuscript, the conclusion is disappointingly brief and fails to bring the key findings back into focus.  It is suggested that this be re-written to bring the Purpose of the manuscript and its subsequent key findings into focus.  This is an important requirement for acceptance. While the Conclusion does hint at what this research progresses there is a need to be more focussed on what new knowledge or understanding is progressed. This should be made clearer in the re-write of the Conclusion.

Thank you very much for your comment. As points 5 and 6 discuss the results and present several reflections and recommendations, we decided to produce a condensed conclusion. However, we agree with the reviewer. As such, we have re-written the Conclusions chapter.

There are 10 keywords listed.  This far too many and the authors need to restrict these to 5.

Thank you for your comment. We followed the guidelines provided by the journal, but it is true: 10 keywords are many keywords. We have restricted these to 5.

The Manuscript need to be thoroughly checked for minor grammatical errors and to ensure that sentences are not too long.  As a general rule word count in any sentence should not exceed 40 words.  For instance, the sentence in lines 56-59 is far to long.  There are other instances of this that need to be addressed.  In addition, Line 78 the word relayed should be ‘relied’.

Thank you very much for your comment. We sent the manuscript to a language expert for an error-free, publication-ready paper.

The sentence in lines 56-59 is a list. Instead of using bullet points, we separate the points through the use of semicolons.

Reviewer 3 Report

The authors try to qualitatively address the factors that might hinder or promote physical activity (PA) in physically disabled people. Theirs was a rich and forthcoming paper with important information for one to reflect on and relay to policy and decision makers and others involved in the PA of disabled people.

I do however have some suggestions and concerns that follow:

  1. There are some typos along the text. Some words or sentence constructions are inadequate. I started to take note but, in the end, I found that there were to many and the text should be corrected by a native speaker. I also found some excerpts of the text to be very informal as I describe below.

  1. I would urge the authors to avoid direct citations (I can understand when they are referring to extracts of the participants discourse but urge against direct citations from other authors). A direct citation would be desirable if pertaining, say, to a historic paper but this does not seem to be the case.

  1. Why do the authors use pseudonyms and not just a number categorization? I understand that one might choose to present such an empathic style. But science should shy away from subjectivity and this is what these extracts portray.

  1. What limitations did the authors find to the ongoing adaptation of the 2 to 1 interview style? Although the authors might have become experts in the final interviews, how did they control for less experience in the first interviews?

  1. Can the authors see any limitation to the fact that some interviews lasted 60 min and others 200min? Could the 200min also account for the participants involvement and rapport obtained? If so, can we envisage less telltale discourse in the 60min participants?

  1. The authors should maybe use a less informal language to describe the methodology. For example, “critical experts” might be used instead of “critical friends” and so forth. Also, sometimes less is more. Such a thorough description of the meetings between experts seems overboard. It is enough to state that there was discussion between the experts…

  1. Why did the authors not insert frequency tables? How might we understand the scope of agreement between the participants? “Largely, the participants” “Most of the participants” and other similar terms are used. However, the reader does not grasp how many exactly. It is important to distinguish an anecdote from the majority.

  1. Again, I do not feel it is necessary to give the participants names. The authors could state “as one participant noted” and distance themselves from a text intended to provoke empathic feelings and move forward to a text intended to share pertinent scientific data.

  1. The authors state again at one point: “Most participants…” 14 vs 13 could refer to “most” just as well as 25 vs 2, I feel that descriptive statistics tables are paramount to support the information given. Up to this point is quite subjective. Moreover, the authors seldom refer to differences between genders in the factor identification. This might also prove rich and insightful. Hence, the insertion of tables where the different factors influencing PA adhesion are systematically described would largely benefit this paper. The way the authors relay the information is somewhat shady in the sense that although the description for category extraction seems logical and adequate, one cannot reproduce such methodological design. More so, I would venture that if a different set of experimenters would get together, they would possibly achieve a different “satisfactory endpoint” and have possibly different recommendations to put forth. In this sense, I recommend that despite the acceptable “methodological bricolage” used for extraction of important categories, from the extraction point on, these should be supported by the use of systematic techniques that allow for reproduction.

  1. It seems that the authors’ explanation of the results entails already a lot of discussion concerning the meaning of those results. In the end, the reader is not sure what are the results exactly and what is the interpretation given by the authors. The authors mix ideas with participant quotes and the literature that gives meaning to it. The authors do a great job in interpreting their findings, but they lose clarity when it comes to identifying the results, the authors’ interpretations and even suggestions as they are all merged together.

  1. The way the authors provide information via extracts of the participants revelations seems somewhat vague and nonspecific. One individual’s needs and difficulties might not mirror another’s as the authors themselves state at a certain point.

  1. The authors state that the perception of stigmatization is present in both men and women. But what frequencies? “Most participants“ is not an indication of quantity and contributes to the current subjectivity of the paper.

  1. Some of the authors’ reflections seem to follow hearsay and, although logical, could fit the results just as well as other theoretical possibilities. For instance, the discourse around the “mirrored body” of gym users seems somewhat abusive.

Despite this hopefully constructive criticism, the authors were able to show how factors can be dynamic and lead to barriers or facilitators, depending on the interaction with each participant. However, I disagree that statistics might conceal these factors. I feel that such qualitative analysis would benefit from the addition of descriptive statistics allowing for (at least) an assortment of the most important factors. Naturally, if one is to make suggestions for changes in Universities and other environments, it might be important to acknowledge the percentage of those that will benefit from these changes. It is not because one does qualitative analyses that one must adhere to all things subjective. I recommend that some of the paper’s subjectivity be withdrawn without extracting the richness which clearly the authors were able to find.          

Author Response

There are some typos along the text. Some words or sentence constructions are inadequate. I started to take note but, in the end, I found that there were to many and the text should be corrected by a native speaker. I also found some excerpts of the text to be very informal as I describe below.

Thank you very much for your comment. We sent the manuscript to a language expert for an error-free, publication-ready paper.

I would urge the authors to avoid direct citations (I can understand when they are referring to extracts of the participants discourse but urge against direct citations from other authors). A direct citation would be desirable if pertaining, say, to a historic paper but this does not seem to be the case.

This is a conventional practice in contemporary qualitative studies. Most of the articles we use in the paper use direct citations.

Why do the authors use pseudonyms and not just a number categorization? I understand that one might choose to present such an empathic style. But science should shy away from subjectivity and this is what these extracts portray.

We use pseudonyms because is coherent with the qualitative nature of this study. This is an accepted (and naturally used) strategy in qualitative reports since it contributes to highlighting the personal and social dimensions of experiences lived by a group of university students with disabilities. As stated in ‘The study’ section, this qualitative study is shaped by a social constructionist paradigm, in which subjectivity is a site of knowledge about the social experiences of people and pseudonyms contribute coherently to communicate the close experiences to the readers.

What limitations did the authors find to the ongoing adaptation of the 2 to 1 interview style? Although the authors might have become experts in the final interviews, how did they control for less experience in the first interviews?

Thank you very much for your comment. We have added some sentences to address this point. Further details on the methodological process used in the two-to-one interviews are indicated in a paper that we mention in the reference list. Since this is an empirical study, sitting with this question would distract readers from the purpose of the paper.

Can the authors see any limitation to the fact that some interviews lasted 60 min and others 200min? Could the 200min also account for the participants involvement and rapport obtained? If so, can we envisage less telltale discourse in the 60min participants?

Qualitative researchers conducting interviews indicate the duration of the interviews in this way, showing difference in terms of duration between the shortest and the longest and agreeing that they are of relatively long duration and have to be reported how long (see Gubrium, Holstein, Marvasti, & McKinney, 2012; Tong, Sainsbury & Craig, 2007). This responds to diverse issues that might be of interest in a methodological paper, such as the amount and balance of researcher and participant talk, the requests for clarification or patterns of turn-taking during the interview (Irvine, 2011). Still, the difference itself does not necessarily indicate methodological flaws.

References:

Gubrium, J.F., Holstein, J.A., Marvasti, A.B. & McKinney, K.D. (2012) The SAGE Handbook of Interview Research: The Complexity of the Craft (2nd ed.). Thousand Oaks, CA: Sage.

Irvine, A. (2011) Duration, Dominance and Depth in Telephone and Face-to-Face Interviews: A Comparative Exploration. International Journal of Qualitative Methods, 10(3), 202-220.

Tong, A., Sainsbury, P.& Craig, J. (2007) Consolidated criteria for reporting qualitative research (COREQ): a 32-item checklist for interviews and focus groups. International Journal for Quality in Health Care, 19 (6), 349 –357.

The authors should maybe use a less informal language to describe the methodology. For example, “critical experts” might be used instead of “critical friends” and so forth. Also, sometimes less is more. Such a thorough description of the meetings between experts seems overboard. It is enough to state that there was discussion between the experts…

“Critical friends”, within the qualitative tradition, becomes the shared and used notion. It is used, for instance, in the reputed international Handbooks on qualitative research (e.g. Denzin and Lincoln, 2000; Denzin and Lincoln, 2005) and also is recognized as part of the rigor when conducting or judging qualitative research (e.g. Smith, B., & McGannon, 2018; una chapter del handbook).

The description of the meetings is slightly developed in this paper to offer rigor and transparency in the methodological section as it is suggested in the previous mentioned literature on qualitative studies.

References:

- Denzin, N.K. & Lincoln, Y.S. (2000). Handbook of qualitative research (2nd ed.). Thousand Oaks, CA: Sage.

- Denzin, N.K. & Lincoln, Y.S. (2005).The Sage Handbook of Qualitative research (3rd ed.) Thousand Oaks, CA: Sage.

- Smith, B., & McGannon, K. R. (2018). Developing rigor in qualitative research: Problems and opportunities within sport and exercise psychology. International review of sport and exercise psychology, 11(1), 101-121.

Why did the authors not insert frequency tables? How might we understand the scope of agreement between the participants? “Largely, the participants” “Most of the participants” and other similar terms are used. However, the reader does not grasp how many exactly. It is important to distinguish an anecdote from the majority.

The focus of this study is not based on the frequency but on the substance appearance of different issues guided by the purpose of the study. According to ‘The Study’ section, this paper precisely highlights the interpretative dimension of facilitators and barriers that is omitted in the quantitative studies based on frequency.

Again, I do not feel it is necessary to give the participants names. The authors could state “as one participant noted” and distance themselves from a text intended to provoke empathic feelings and move forward to a text intended to share pertinent scientific data.

According to the specialized literature, the researchers and participants are ‘present’ in the text of a qualitative report (e.g. Sparkes, A. (2002). Telling tales in sport and physical activity: A qualitative journey. Human Kinetics Publishers.

The authors state again at one point: “Most participants…” 14 vs 13 could refer to “most” just as well as 25 vs 2, I feel that descriptive statistics tables are paramount to support the information given. Up to this point is quite subjective. Moreover, the authors seldom refer to differences between genders in the factor identification. This might also prove rich and insightful. Hence, the insertion of tables where the different factors influencing PA adhesion are systematically described would largely benefit this paper. The way the authors relay the information is somewhat shady in the sense that although the description for category extraction seems logical and adequate, one cannot reproduce such methodological design. More so, I would venture that if a different set of experimenters would get together, they would possibly achieve a different “satisfactory endpoint” and have possibly different recommendations to put forth. In this sense, I recommend that despite the acceptable “methodological bricolage” used for extraction of important categories, from the extraction point on, these should be supported by the use of systematic techniques that allow for reproduction.

The reviewer should consider this is a qualitative study as indicated in the text since the beginning of the paper. Qualitative interpretative researchers are not experimenters and they do not use descriptive statistics in their procedures. They do not shy away from subjectivity. They do not use methodological designs that produce reproducible research.

It seems that the authors’ explanation of the results entails already a lot of discussion concerning the meaning of those results. In the end, the reader is not sure what are the results exactly and what is the interpretation given by the authors. The authors mix ideas with participant quotes and the literature that gives meaning to it. The authors do a great job in interpreting their findings, but they lose clarity when it comes to identifying the results, the authors’ interpretations and even suggestions as they are all merged together.

The reviewer seems to object to the fact that ‘The authors mix ideas with participant quotes and the literature that gives meaning to it’. However, this is what qualitative analysis is supposed to be. Please note that we make explicit our epistemological position in the paper (p. 3):

  • The present qualitative study was underpinned by an interpretivist paradigm [12], which assumes a relativist ontology (reality is multiple and mind-dependent) and a subjectivist epistemology (knowledge is socially constructed and subjective). More concretely, we adopted a thick interpretative (as opposed to a descriptive) position [13]. Such a position allows researchers to give meaning to participants' experiences beyond that which the participants may be able or willing to attribute to it. As Willig [13] elaborated, emphasizing on interpretation means 'abandoning any truth claims regarding the analytic insights produced, arguing instead that what is being offered is the researcher's reading of the data' (p. 9). To cool down any possible reaction against 'bias', see -for example- the work of Ponterotto [12], Sparkes and Smith [14] or Smith and McGannon [15], which explains why interpretivist research cannot and should not be judged through post-positivist logics.

The way the authors provide information via extracts of the participants revelations seems somewhat vague and nonspecific. One individual’s needs and difficulties might not mirror another’s as the authors themselves state at a certain point.

Our study does not intend to generate statistical-probabilistic generalisations. Please see this article: Smith, B. (2018). Generalizability in qualitative research: Misunderstandings, opportunities and recommendations for the sport and exercise sciences. Qualitative research in sport, exercise and health, 10(1), 137-149.

Despite this hopefully constructive criticism, the authors were able to show how factors can be dynamic and lead to barriers or facilitators, depending on the interaction with each participant. However, I disagree that statistics might conceal these factors. I feel that such qualitative analysis would benefit from the addition of descriptive statistics allowing for (at least) an assortment of the most important factors. 

We appreciate your constructive criticism. However, qualitative research is not subject to the same philosophical assumptions of quantitative research. We produce quantitative research, and we follow the appropriate conventions to do so. We have produced mixed-methods research before. This time, however, we have produced a qualitative study which stands alone, as many other qualitative studies do.